# Interpretable and Interactive Deep Multiple Instance Learning for Dental Caries Classification in Bitewing X-rays

**Benjamin Bergner**[1,5]                                    BENJAMIN.BERGNER@HPI.DE
**Csaba Rohrer**[2]                                          CSABA.ROHRER@CHARITE.DE
**Aiham Taleb**[1]                                           AIHAM.TALEB@HPI.DE
**Martha Duchrau**[2]                                        MARTHA.DUCHRAU@CHARITE.DE
**Guilherme De Leon**[3]                          GUILHERME@CONTRASTERADIOLOGIA.COM
**Jonas Almeida Rodrigues**[4]                               JORODRIGUES@UFRGS.BR
**Falk Schwendicke**[2]                                  FALK.SCHWENDICKE@CHARITE.DE
**Joachim Krois**[2]                                       JOACHIM.KROIS@CHARITE.DE
**Christoph Lippert**[1,5]                              CHRISTOPH.LIPPERT@HPI.DE

[1] *Digital Health & Machine Learning, Hasso Plattner Institute, University of Potsdam, Germany*

[2] *Department of Oral Diagnostics, Digital Health and Health Services Research, Charité - Universitätsmedizin Berlin, Germany*

[3] *Contraste Radiologia Odontológica, Blumenau, Brasil*

[4] *Universidade Federal do Rio Grande do Sul – UFRGS, School of Dentistry, Department of Surgery and Orthopedics, Porto Alegre, RS, Brazil.*

[5] *Hasso Plattner Institute for Digital Health at Mount Sinai, Icahn School of Medicine at Mount Sinai, NYC, USA*

**Editors:** Under Review for MIDL 2022

## Abstract

We propose a simple and efficient image classification architecture based on deep multiple instance learning, and apply it to the challenging task of caries detection in dental radiographs. Technically, our approach contributes in two ways: First, it outputs a heatmap of local patch classification probabilities despite being trained with weak image-level labels. Second, it is amenable to learning from segmentation labels to guide training. In contrast to existing methods, the human user can faithfully interpret predictions and interact with the model to decide which regions to attend to. Experiments are conducted on a large clinical dataset of ∼38k bitewings (∼316k teeth), where we achieve competitive performance compared to various baselines. When guided by an external caries segmentation model, a significant improvement in classification and localization performance is observed.

**Keywords:** dental deep learning, MIL, interpretability, interactive learning

## 1. Introduction

Dental caries is the most prevalent health disease, affecting more than three billion people worldwide (Kassebaum et al., 2017). For diagnosis, clinicians commonly analyze bitewing radiographs (BWRs), which show the maxillary and mandibular teeth of one side of the jaw. However, the assessment of caries in bitewings is associated with low detection rates. For example, Schwendicke et al. (2015) reported domain expert-level sensitivity of only 24% (21%-26%, 95% CI) for the detection of both initial and advanced carious lesions.

The challenging nature of caries detection and the growing quantity of dental data motivate the use of deep learning techniques for this task. In order to support dentists, such models must overcome various technical challenges, as follows: **(1)** Diagnosing caries is a low signal-to-noise ratio problem. That is, lesions may occupy only few pixels in the image. Standard convolutional neural networks (CNN) have been shown to struggle in this setting (Pawlowski et al., 2020). In contrast, models that use attention are designed to focus on important regions while ignoring the prevalent background (Katharopoulos and Fleuret, 2019). **(2)** Training a caries classification model with image-level labels is a multiple instance learning (MIL) problem (Dietterich et al., 1997). That is, an image is considered positive if at least one carious lesion is present and negative if and only if no lesion is present. In this context, an image is described as a bag of image region features called instances. Labels are only available for bags but not individual instances; see Carbonneau et al. (2018) for an introduction. **(3)** BWRs contain multiple teeth, and each may be affected by caries. However, classification outputs are restricted to a single probability score and thus lack interpretability (Zhang and Zhu, 2018). A supporting model should indicate where each lesion is located so that its correctness can be verified. **(4)** Optimal decision support is receptive to feedback (Holzinger, 2016). Beyond only outputting information about the occurrence of caries (learned from weak labels), a dentist or teacher (Hinton et al., 2015) could interact with the model by providing strong labels (such as segmentation masks) to improve performance.

We present **E**mbedding **M**ultiple **I**nstance **L**earning (EMIL), which is an interpretable and interactive method that fulfills above considerations. EMIL extracts patches from a spatial embedding resulting from any CNN. Each patch may show caries and is classified individually, and all predictions together form a heatmap of local probabilities, notably without access to patch labels. An attention mechanism weighs local predictions and aggregates them into a global image-level prediction. Besides standard classification, the method enables (but does not rely on) the inclusion of dense labels. Although EMIL adds important capabilities for the present use case, classification of dental caries, it is a simple adaptation to common CNNs with low computational cost that translates to other diagnosis tasks. We evaluate performance and interpretability using a large clinical bitewing dataset for image- and tooth-level classification, and show the positive impact of including strong tooth and caries labels. Our code is available at: https://github.com/benbergner/emil.

## 2. Related Work

### 2.1. Caries prediction models

Recently, several caries prediction models have been published. Tripathi et al. (2019) used a genetic algorithm on 800 BWRs and reported an accuracy of 95.4%. Srivastava et al. (2017) trained a 100+ layer CNN on 2,500 BWRs and reported an F-score of 70%. Megalan Leo and Kalpalatha Reddy (2020) trained a CNN on 418 cropped teeth from 120 BWRs and achieved an accuracy of 87.6%. Kumar and Srivastava (2018) proposed an incremental learning approach and trained a U-Net on 6,000 BWRs, which yielded an F-score of 61%. Cantu et al. (2020) trained a U-Net on 3,686 BWRs and reported tooth-level accuracy and F-score of 80% and 73%, respectively. Bayraktar and Ayan (2021) trained YOLO on 800 bitewings and reported an AUC score of 87%.

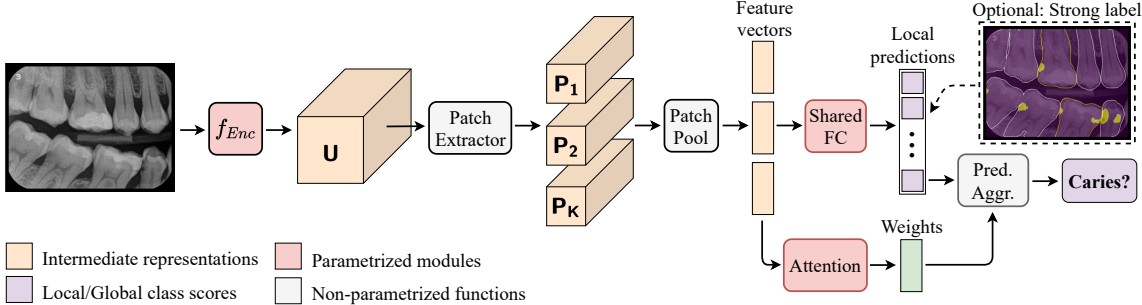

Figure 1: EMIL classification architecture.

## 2.2. Deep Multiple Instance Learning

MIL is commonly used for the classification of microscopic images in which, e.g., a single cancer cell positively labels a bag (Kraus et al., 2016; Sudharshan et al., 2019). MIL has also recently been applied in radiology, e.g. Han et al. (2020) screened chest CTs for COVID-19 and Zhou et al. (2018) detected diabetic retinopathy in retinal images. To the best of our knowledge, this is the first application of MIL to the field of dental radiology.

Instance representations are commonly created from patches extracted from the input image (Xu et al., 2014), but can also be extracted from a CNN embedding (Pawlowski et al., 2020; Dosovitskiy et al., 2021). Furthermore, one can distinguish between approaches predicting at the instance (Wu et al., 2015; Campanella et al., 2018) or bag-level (Wang et al., 2018; Ilse et al., 2018). Our approach combines the extraction of instances as overlapping patches from a CNN embedding with the classification of individual instances that are aggregated with a constrained form of attention-based MIL pooling.

## 3. Method

Below, we describe our proposed model, the creation of a local prediction heatmap, and a method to incorporate strong labels. A schematic of the architecture is shown in Figure 1.

### 3.1. Patch extraction and classification

We consider an image $\mathbf{X} \in \mathbb{R}^{H_X \times W_X \times C_X}$ as input, with $H_X$, $W_X$, $C_X$ being height, width and number of channels, and assign a binary label $y \in \{0, 1\}$. First, a(ny) convolutional backbone computes a feature map $\mathbf{U} \in \mathbb{R}^{H_U \times W_U \times C_U}$:

$$\mathbf{U} = f_{\text{Enc}}(\mathbf{X}). \tag{1}$$

Then, $K$ patches $\mathbf{P} \in \mathbb{R}^{K \times H_P \times W_P \times C_U}$ are extracted from $\mathbf{U}$, with $H_P \leq H_U, W_P \leq W_U$. For this purpose, a sliding window is used with kernel size $(H_P, W_P)$ and stride $(H_S, W_S)$. Each patch is spatially pooled, resulting in a feature matrix $\boldsymbol{P} \in \mathbb{R}^{K \times C_U}$. We use average pooling, which is most prevalent in image classification (Lin et al., 2014):

$$\boldsymbol{P}_k = \frac{1}{H_P W_P} \sum_{h=1}^{H_P} \sum_{w=1}^{W_P} \mathbf{P}_{k,h,w}. \tag{2}$$

Both patch extraction and pooling are implemented by an ordinary local pooling operation. Each patch may show a carious tooth region and is thus classified independently by a shared fully-connected layer parametrized by $\boldsymbol{o} \in \mathbb{R}^{C_U}$. This is followed by a sigmoid operator, which outputs classifications $\tilde{\boldsymbol{y}} \in \mathbb{R}^K$ holding class probabilities for each patch:

$$\tilde{y}_k = \sigma((\boldsymbol{Po})_k), \qquad\qquad k = 1 \ldots K. \qquad\qquad (3)$$

## 3.2. Patch weighting and aggregation

We use a patch weight vector $\boldsymbol{w} \in \mathbb{R}^{K \times 1}$ to focus on carious lesions while neglecting background and non-caries tooth regions. The image-level prediction $\hat{y}$ is computed as:

$$\hat{y} = \frac{\sum_k^K w_k \tilde{y}_k}{\max\left(\sum_k^K w_k, K_{min}\right)}. \qquad\qquad (4)$$

The denominator ensures that at least $K_{min}$ patches are attended to, and provides a way to include prior knowledge about the target's size. For caries classification, a single positive patch should lead to a positive prediction, so we set $K_{min} = 1$ (see Appendix D for more details). The weight of each patch is determined by its own local representation. We use a variant of the gated attention mechanism (Ilse et al., 2018), which is a two-branch multilayer perceptron parametrized by $\boldsymbol{A} \in \mathbb{R}^{C_U \times D}$, $\boldsymbol{B} \in \mathbb{R}^{C_U \times D}$ and $\boldsymbol{c} \in \mathbb{R}^{D \times 1}$, with $D$ hidden nodes:

$$\boldsymbol{w} = \sigma\Big(\big(\tanh(\boldsymbol{PA}) \odot \sigma(\boldsymbol{PB})\big)\boldsymbol{c}\Big). \qquad\qquad (5)$$

Compared to the original formulation using softmax, we employ sigmoid as outer function, and normalize Eq. 4 accordingly. This makes the weights independent of each other and allows to ignore all patches, which is useful for classifying the negative class.

## 3.3. Interpretability

A heatmap $\boldsymbol{M}^{\tilde{\boldsymbol{y}}}$ is constructed with each element corresponding to a local prediction $\tilde{y}_k$. For visualization, the heatmap is interpolated and superimposed on the input image. Similarly, another heatmap $\boldsymbol{M}^{\boldsymbol{w}}$ is built from patch weights $\boldsymbol{w}$. While $\boldsymbol{M}^{\tilde{\boldsymbol{y}}}$ shows local predictions, $\boldsymbol{M}^{\boldsymbol{w}}$ indicates which areas are considered for global classification. Note that the locations in $\boldsymbol{M}^{\tilde{\boldsymbol{y}}}$ and $\boldsymbol{M}^{\boldsymbol{w}}$ can be interpreted as probabilities, a property that attribution methods lack. The probability for any group of patches (e.g., a tooth) can be calculated with Eq. 4 by updating patch indices in both sums. Furthermore, note that EMIL is optimized for faithfulness (Alvarez Melis and Jaakkola, 2018). That is, one can tell exactly by how much $\hat{y}$ changes when removing any patch $i$, which is $-\frac{w_i \tilde{y}_i}{K_{min}}$ if $\sum_k w_k - w_i < K_{min}$ and 0 otherwise. For example, if $K_{min} = 1$ and 2 caries patches are present, $\hat{y}$ shouldn't change by removing one caries patch, which aligns with the standard MIL assumption (Foulds and Frank, 2010).

## 3.4. Interactivity

Optionally, learning can be guided by providing additional labels, such as segmentation masks. For example, a dentist could interactively correct errors/biases; while a data scientist might want to incorporate dense (and expensive) labels for a subset of the data. To create

patch-wise labels $\boldsymbol{y} \in \mathbb{R}^K$, a downscaled binary annotation mask is max-pooled with kernel size/stride from Sect. 3.1, and vectorized. Then, to compute compound loss $\mathcal{L}$ for an image, both patch and image-level cross-entropy losses $\boldsymbol{\ell} = [\mathcal{L}_{image}, \mathcal{L}_{patch}]$ are weighted and added:

$$\mathcal{L}_{image} = -\big(y \log(\hat{y}) + (1 - y) \log(1 - \hat{y})\big), \tag{6}$$

$$\mathcal{L}_{patch} = -\frac{1}{K} \sum_{k=1}^{K} \big(y_k \log(\tilde{y}_k) + (1 - y_k) \log(1 - \tilde{y}_k)\big), \tag{7}$$

$$\mathcal{L} = \sum_i \alpha_i \ell_i, \quad \alpha_i = \mathrm{const}\left(\frac{\max_j \ell_j}{\ell_i}\right). \tag{8}$$

Due to class imbalance in caries masks, the network easily fits the background class and we observe that $\mathcal{L}_{patch} \ll \mathcal{L}_{image}$. Thus, $\mathcal{L}_{image}$ dominates the compound loss and diminishes the benefit of strong labels. To mitigate this problem, coefficient $\alpha$ is introduced to dynamically scale each partial loss to the magnitude of the largest one. Note that $\alpha$ is transformed into a constant so that the partial losses are detached from the computational graph.

## 4. Experiments

Below, we describe the experiments and answer the following research questions: (1) How well can EMIL predict caries in BWRs and cropped tooth images? (2) Can it highlight caries and provide clinical insight? (3) To what extent do strong labels improve performance?

### 4.1. Dataset

The dataset stems from three dental clinics in Brazil specialized in radiographic and tomographic examinations. The dataset consists of 38,174 BWRs (corresponding to 316,388 cropped tooth images) taken between 2018 and 2021 from 9,780 patients with a mean (sd) age of 34 (14) years. Tooth-level caries labels were extracted from electronic health records (EHRs) that summarize a patient's dental status. Next to these EHR-based ground truth labels, which are associated with uncertainties and biases (Gianfrancesco et al., 2018), a random sample of 355 BWRs was drawn, and annotated with caries masks by 3 experienced dentists, yielding 254 positive and 101 negative cases. These annotations were reviewed by a senior dentist with +13 years of experience to resolve conflicts and establish a test set.

### 4.2. Experimental setup

We consider caries classification on BWR and tooth level and use stratified 5-fold cross-validation with non-overlapping patients for training and hyperparameter tuning. Due to class imbalance, the balanced accuracy is used as stopping criterion. In the tooth-level task, both class terms in $\mathcal{L}_{image}$ are weighted by the inverse class frequency to account for class imbalance. Results are reported on the hold-out test set as average of the 5 resulting models with 95% CI. Binary masks from two teacher models are used to simulate interactivity: (1) a tooth instance-segmentation model (🦷, unpublished) pointing at affected teeth and (2) a caries segmentation model (🦷, Cantu et al. (2020)). Note that these models are subject to errors and do not replace class labels, but only guide training; if a segmentation contradicts the classification label, it is discarded. More training details are described in Appendix C.

Table 1: Caries classification results with 95% CI and computational comparison.

| | | Bal. Acc. | F-score | Sens. | Spec. | Time [ms] | RAM [GB] |
|---|---|---|---|---|---|---|---|
| **Bitewing (512x672)** | ResNet-18 | $73.31 \pm 2.65$ | $75.03 \pm 0.87$ | $64.25 \pm 2.25$ | $\mathbf{82.38 \pm 7.08}$ | 44 | 2.48 |
| | ResNet-50 | $71.15 \pm 3.68$ | $75.90 \pm 0.85$ | $67.24 \pm 3.12$ | $75.05 \pm 10.19$ | 143 | 9.19 |
| | DeepMIL-32 | $67.28 \pm 1.07$ | $75.17 \pm 3.11$ | $68.43 \pm 5.89$ | $66.14 \pm 6.64$ | 256 | 9.43 |
| | DeepMIL-128 | $70.68 \pm 2.92$ | $73.08 \pm 5.85$ | $62.76 \pm 9.50$ | $78.61 \pm 7.95$ | 133 | 7.51 |
| | ViT | $72.92 \pm 3.67$ | $74.35 \pm 3.90$ | $63.46 \pm 7.18$ | $82.38 \pm 11.51$ | 50 | 3.40 |
| | EMIL | $73.64 \pm 1.77$ | $\mathbf{77.88 \pm 2.11}$ | $\mathbf{69.45 \pm 4.43}$ | $77.82 \pm 6.65$ | 45 | 2.48 |
| | ResNet-18 + 🦷 | $74.10 \pm 4.43$ | $73.19 \pm 9.51$ | $61.26 \pm 12.34$ | $\mathbf{86.93 \pm 5.32}$ | 47 | 2.48 |
| | ResNet-18 + 🦷 | $75.40 \pm 2.17$ | $77.32 \pm 3.18$ | $67.24 \pm 5.95$ | $83.56 \pm 7.41$ | 47 | 2.48 |
| | Y-Net + 🦷 | $75.78 \pm 1.41$ | $77.13 \pm 3.07$ | $66.61 \pm 5.27$ | $84.95 \pm 4.38$ | 318 | 23.54 |
| | Y-Net + 🦷 | $75.90 \pm 2.47$ | $77.56 \pm 1.74$ | $67.24 \pm 2.14$ | $84.55 \pm 4.31$ | 318 | 23.54 |
| | EMIL + 🦷 | $74.69 \pm 2.12$ | $77.79 \pm 3.07$ | $68.58 \pm 4.82$ | $80.79 \pm 3.64$ | 48 | 2.48 |
| | EMIL + 🦷 | $\mathbf{76.64 \pm 1.50}$ | $\mathbf{79.52 \pm 3.48}$ | $\mathbf{70.71 \pm 6.76}$ | $82.57 \pm 7.66$ | 48 | 2.48 |
| | EHR GT | 80.90 | 83.11 | 74.80 | 87.00 | - | - |
| **Tooth (384x384)** | ResNet-18 | $75.13 \pm 0.72$ | $65.40 \pm 0.97$ | $62.39 \pm 2.66$ | $87.88 \pm 1.85$ | 24 | 1.17 |
| | ResNet-50 | $74.72 \pm 1.05$ | $64.88 \pm 1.34$ | $60.57 \pm 4.41$ | $88.88 \pm 2.44$ | 68 | 4.14 |
| | DeepMIL-32 | $70.04 \pm 0.80$ | $58.03 \pm 1.13$ | $59.75 \pm 4.94$ | $80.33 \pm 4.04$ | 105 | 4.03 |
| | DeepMIL-128 | $74.52 \pm 0.72$ | $64.60 \pm 0.98$ | $60.16 \pm 2.54$ | $88.88 \pm 1.45$ | 51 | 2.92 |
| | ViT | $74.52 \pm 1.01$ | $64.73 \pm 1.38$ | $58.54 \pm 4.12$ | $\mathbf{90.49 \pm 2.53}$ | 29 | 1.56 |
| | EMIL | $\mathbf{75.67 \pm 1.74}$ | $\mathbf{66.02 \pm 2.25}$ | $\mathbf{64.16 \pm 5.26}$ | $87.17 \pm 2.14$ | 25 | 1.18 |
| | ResNet-18 + 🦷 | $74.99 \pm 1.32$ | $65.53 \pm 1.90$ | $58.57 \pm 3.93$ | $\mathbf{91.41 \pm 2.06}$ | 26 | 1.17 |
| | Y-Net + 🦷 | $\mathbf{76.40 \pm 1.13}$ | $\mathbf{67.50 \pm 1.49}$ | $62.21 \pm 3.83$ | $90.59 \pm 2.10$ | 148 | 10.25 |
| | EMIL + 🦷 | $76.14 \pm 1.29$ | $67.01 \pm 1.86$ | $\mathbf{62.68 \pm 4.37}$ | $89.59 \pm 3.12$ | 26 | 1.18 |
| | EHR GT | 70.03 | 57.44 | 44.27 | 95.79 | - | - |

### 4.3. Baselines

Several baselines are used to show competitive performance. ResNet-18 (He et al., 2016) serves as backbone for all methods. EMIL makes only few changes to the default CNN, so we also employ ResNet-18 as a baseline. In order to study the effect of embedding-based patch extraction, we compare to DeepMIL (Ilse et al., 2018), which operates on patches cropped from the input image. As patch sizes, 32 and 128 px with 50% overlap are used.

To show the effect of our patch weighting and aggregation approach, the attention mechanism is replaced by the max operator, which is common in instance-based MIL (Amores, 2013). However, this did not fit the training data. A more powerful baseline is the hybrid version of the Vision Transformer (ViT) (Dosovitskiy et al., 2021) with a single encoding block. As in EMIL, patches are extracted from the output of the last conv layer, and we found large overlapping patches to be beneficial. The hybrid base and pure attention versions were not included because they performed worse or did not fit the training data.

For the evaluation of the interactive settings, a simple baseline consists in attaching a 1x1 conv layer (Lin et al., 2014) with a single output channel, kernel and stride of 1 and zero padding, to the last ResNet-18 encoder block to output a segmentation map. As a stronger baseline, we adapt Y-Net (Mehta et al., 2018), which consists of a U-Net (with a ResNet-18 encoder) and a standard output layer attached to the last encoding block for classification. In all interactive settings, the same loss functions are used (Eq. 6-8). See also Appendix B.

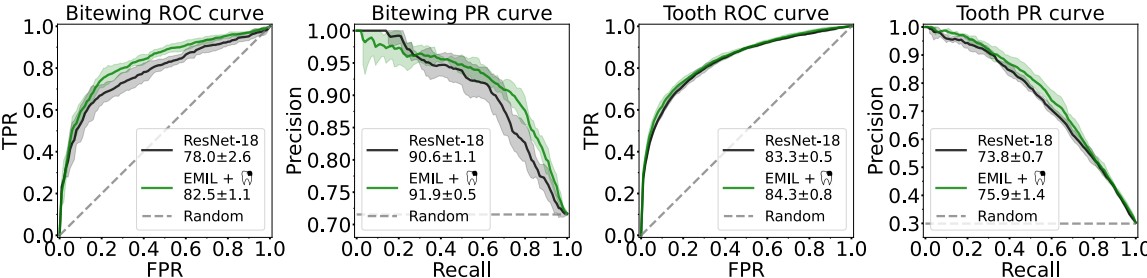

Figure 2: ROC and PR curves with AUC values and CIs for bitewing and tooth datasets.

## 4.4. Classification results

Table 1 and Fig. 2 summarize the results. In BWRs, EMIL shows highest accuracy, F-score and sensitivity across settings (in bold). In contrast, increasing capacity (ResNet-50) and complex self-attention based aggregators (ViT) do not improve performance. Furthermore, DeepMIL-32/128 show lower accuracy indicating that context beyond patch borders is crucial. The use of tooth/caries masks increases mean performance. In particular, EMIL + 🦷 exhibits higher scores across metrics compared to a non-guided model and has a significantly higher F-score and AUROC than ResNet-18. These trends continue in the tooth-level data, although improvements of guided models are smaller. This is expected because the signal-to-noise ratio is higher than in BWRs. We also report results for clinical labels (EHR GT) on which all models are based. The summary metrics (Bal. Acc., F-score) are higher for BWRs, possibly because errors at the tooth-level may still lead to true positives. Intriguingly, all tooth-level models show a higher accuracy/F-score than EHR GT. This suggests that salient patterns are learned that clinicians missed (or did not report) and may result from the fact that mislabeled false positives have less weight in the loss function. Table 1 also reports average runtimes per iteration and peak memory usage for a realistic batch size of 16. EMIL is nearly as efficient as its underlying backbone, and up to 6.6× faster than Y-Net, while consuming up to 9.5× less memory at similar/better mean performance.

## 4.5. Evaluation of interpretability

Fig. 3 shows a positive BWR and EMIL heatmaps. $M^{\tilde{y}}$ is sensitive and detects all lesions, while $M^w$ is precise and focuses on the most discriminative regions. A colorbar indicates local class probabilities and attention values, respectively. Fig. 4 adds a qualitative visualization comparison. Attribution methods such as saliency maps (Simonyan et al., 2014), Grad-CAM (Selvaraju et al., 2017) or occlusion maps (Zeiler and Fergus, 2014) (ResNet), as well as DeepMIL, are sensitive to positive cases (rows 1-3) but not precise. Moreover, these methods do not ignore the negative class (row 4), and false negatives are accompanied by false positive visualizations (row 5). This is resolved in Y-Net and EMIL ($M^{\tilde{y}}$), and caries may be highlighted although the activation is too low to cross the classification threshold (e.g., row 5, column 8). Table 2 adds a quantitative comparison where the overlap of ground truth and heatmaps is computed as Intersection over Union (IoU, in %) for the positive class. For a fair comparison, we follow Viviano et al. (2021) and set the topmost pixels of each map to 1, so that the total area equals the respective ground truth. When considering all confidences (IoU@0), Saliency and EMIL localize best. In the interactive set-

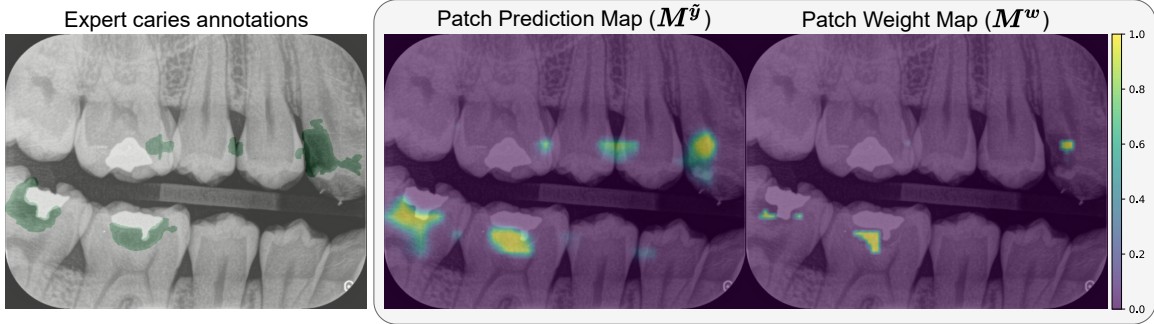

Figure 3: Visualization on a test image. EMIL is trained without expert annotations.

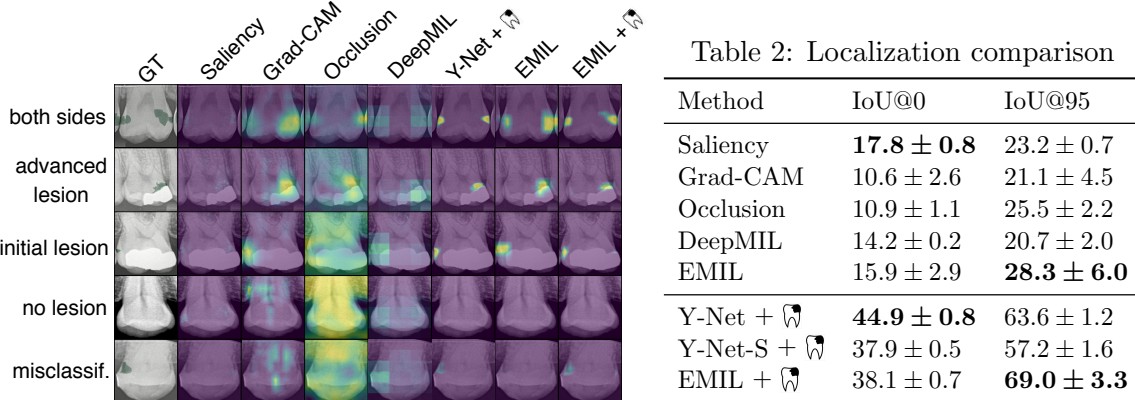

Figure 4: Tooth visualizations of different methods

Table 2: Localization comparison

| Method | IoU@0 | IoU@95 |
|---|---|---|
| Saliency | $\mathbf{17.8 \pm 0.8}$ | $23.2 \pm 0.7$ |
| Grad-CAM | $10.6 \pm 2.6$ | $21.1 \pm 4.5$ |
| Occlusion | $10.9 \pm 1.1$ | $25.5 \pm 2.2$ |
| DeepMIL | $14.2 \pm 0.2$ | $20.7 \pm 2.0$ |
| EMIL | $15.9 \pm 2.9$ | $\mathbf{28.3 \pm 6.0}$ |
| Y-Net + 🦷 | $\mathbf{44.9 \pm 0.8}$ | $63.6 \pm 1.2$ |
| Y-Net-S + 🦷 | $37.9 \pm 0.5$ | $57.2 \pm 1.6$ |
| EMIL + 🦷 | $38.1 \pm 0.7$ | $\mathbf{69.0 \pm 3.3}$ |

tings, scores improve significantly. Y-Net shows high IoU due to its parametrized decoder, which outputs high-res masks. When replacing the decoder by a bilinear upsampler (factor 4) + 1x1 conv output layer (Y-Net-S), results are comparable to EMIL. We also conduct an experiment where only confident predictions ($\hat{y} \geq 0.95$) are retained. The localization performance increases most in EMIL models, by 12.4 and 30.9 percentage points.

## 5. Discussion

We presented two caries classifiers for bitewing and tooth images. The former indicates the general presence of caries in the dentition (with high PR AUC), while the latter may support diagnosis. One limitation is that training is performed with EHRs, which makes the labels error-prone. However, our dataset is much larger than related work, and relabeling at scale is impractical. Yet, the tooth-level model shows higher sensitivity than clinicians ($62.68 \pm 4.37$ vs. 44.27), suggesting that more lesions can be found and treated in practice. The heatmaps are a useful tool to see on what grounds a prediction is made, and to estimate caries severity. Furthermore, we showed that strong labels pointing at relevant regions improve classification and localization, opening up ways to integrate the user into the training process. Technically, our approach may serve further computer-aided diagnosis applications in radiology, where trust and the ability to integrate human knowledge are critical.

## Acknowledgments

This project has received funding by the German Ministry of Research and Education (BMBF) in the projects SyReal (project number 01|S21069A) and KI-LAB-ITSE (project number 01|S19066). We would also like to thank Matthias Kirchler for fruitful discussions about the architecture and Florian Sold for managing the HPC.

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

## Appendix A. Tooth-level classification results

Table 3 shows classification results and prevalence for different tooth and caries types for the guided EMIL model. In terms of tooth types, the model performs significantly worse for canines, which can be explained by the low prevalence in the data. A higher average F-score is observed for premolars compared to molars. The former may be easier to detect because they appear centrally in the bitewing and are unlikely to be partially cut out of the image. In secondary/recurrent caries, the average sensitivity is higher than in primary/initial caries. One possible reason for this result is that such lesions are adjacent to restorations which are radiopaque, sharply demarcated and easy to detect. In addition, secondary caries can spread more quickly because it no longer has to penetrate the hard enamel, but can quickly reach the softer interior of the tooth.

Table 3: Classification results for different tooth and caries types for EMIL + 🦷 .

|  | Bal. Acc. | F-score | Sens. | Spec. | Preval. |
|---|---|---|---|---|---|
| Canine | $67.48 \pm 4.80$ | $45.47 \pm 7.14$ | $42.55 \pm 12.25$ | $92.42 \pm 4.14$ | 11.0 |
| Premolar | $76.88 \pm 1.21$ | $69.08 \pm 1.69$ | $64.11 \pm 4.52$ | $89.66 \pm 3.25$ | 41.5 |
| Molar | $76.18 \pm 1.49$ | $67.50 \pm 2.15$ | $63.74 \pm 4.17$ | $88.63 \pm 2.86$ | 47.5 |
| Primary | $76.63 \pm 0.98$ | $61.99 \pm 2.17$ | $63.68 \pm 3.58$ | $89.59 \pm 3.12$ | 17.4 |
| Secondary | $77.67 \pm 2.11$ | $62.25 \pm 3.21$ | $65.74 \pm 5.70$ | $89.59 \pm 3.12$ | 16.0 |

## Appendix B. Conceptual architecture comparison

In Table 4, we revisit the baselines used in the main paper and make a conceptual comparison. EMIL and ViT (hybrid) extract patches from a CNN embedding, while DeepMIL extracts patches from the input image. The embedding approach has the advantage that the context is large due to the growing receptive field resulting from the sequence of convolutional layers. The MIL literature (Carbonneau et al., 2018; Amores, 2013) distinguishes between two different types of inputs for the classification function. The first type is the bag representation, which is calculated by aggregating instances using a MIL pooling operation (such as mean or attention). The second type used by EMIL is individual instances that are classified before aggregation. Standard CNNs instead use a global embedding, without distinguishing between instances and bags. There are also different assumptions about when a bag is considered positive (Foulds and Frank, 2010). The most common is the standard assumption (any positive instance → bag positive; no positive instance → bag negative). In the weighted collective assumption (used by DeepMIL), all instances are considered in a weighted manner to infer the class of the bag. EMIL uses both the weighted collective and the threshold-based assumption, where a minimum number of patches ($K_{min}$) must be positive for the bag to be classified as positive.

Standard CNNs are black boxes that require post-hoc attribution methods to give insights about their predictions (in ViT, attention maps can be visualized as well). The drawback of such methods is that they are not optimized for faithfulness (Adebayo et al.,

2018; Rudin, 2019) and cannot explain the negative class (see Fig. 4, row 4). Y-Net is interpretable through its decoder but requires segmentation labels and does not use the decoder output for classification. DeepMIL uses attention, but weights need to sum to 1, which is unintuitive for the negative class. EMIL uses both attention weights and patch probabilities to create faithful explanations for a prediction. Regarding interactive learning, standard CNNs are trained with classification labels but cannot learn from dense labels. Y-Net and EMIL are both able to learn from dense labels, but EMIL does it efficiently.

Table 4: Conceptual comparison

| Method | Patch extraction | Classifier input | MIL assumption | Interpretability | Interactivity |
|--------|------------------|------------------|----------------|------------------|---------------|
| ResNet | ✗ | Global embedding | ✗ | Post-hoc | ✗ |
| ViT | Embedding | Bag Representation | ✗ | Post-hoc | ✗ |
| Y-Net | ✗ | Global embedding | ✗ | Decoder | ✓ |
| DeepMIL | Input image | Bag Representation | Weighted collective | Patch weights | ✗ |
| EMIL | Embedding | Instance Representation | Threshold-based + Weighted collective | Patch weights + Patch probabilities | ✓ |

## Appendix C. Implementation and training details

### C.1. Baseline implementations

The ResNet-18 backbone is based on the original Pytorch implementation[1]. For DeepMIL, the original implementation of Ilse et al. (2018)[2] is used. For ViT, we adapt the `vit-pytorch` repository[3] and found that a minimal hybrid version using a single transformer encoder block, with 8 heads (each 64-dimensional), works best. Both patch representations and inner MLP layers are 128-dimensional. No dropout is used, and all patch representations are averaged before the MLP head. We employ the same approach as in EMIL and create overlapping patches with a large kernel size of 5 and a stride of 1. For Y-Net, we adapt the residual U-Net implementation of the `ResUnet` repository[4], where we add a fully-connected output layer to the bottleneck and learn the upsampling in the decoder. For saliency maps, occlusion sensitivity and Grad-CAM, we use the Captum library (Kokhlikyan et al., 2020).

### C.2. Preprocessing

The dataset consists of 38,174 bitewings, which corresponds to 316,388 teeth. To prepare the data, we make use of the following exclusion criteria. BWRs are excluded if a single tooth is located on the wrong jaw side or if all carious lesions occur in incisors. Similarly, a tooth image is excluded if it shows an incisor or if the tooth is located on the wrong jaw side. We remove all images from the test set, as well as other images from test set patients. After applying these filters, 36,676 bitewings (26,393 with caries) and 274,877 teeth (59,859 with caries) remain for training. The test set consists of 355 BWRs, 254

---

1. https://github.com/pytorch/vision/blob/main/torchvision/models/resnet.py

2. https://github.com/AMLab-Amsterdam/AttentionDeepMIL

3. https://github.com/lucidrains/vit-pytorch

4. https://github.com/rishikksh20/ResUnet

positive, 101 negative. This corresponds to 2,938 tooth images, 879 positive, 2,059 negative. Bitewing images are resized to 512x672 pixels, which preserves the prevalent height to width ratio. Tooth images are cropped with a 50-pixel padding on each side, and then resized to 384x384 pixels. Intensities are normalized in the range [-1,1]. No other augmentations (such as rotations, translations, contrast enhancements, AutoAugment) are used, as no improvement was observed.

### C.3. Optimization

We use the Adam optimizer (Kingma and Ba, 2015) with a learning rate of 0.001, $\beta_1 = 0.9$, $\beta_2 = 0.999$ and no weight decay. In each fold, we train for 20-30 epochs depending on the training progress of the respective methods. We observed that training in the interactive settings is faster, which is expected as the segmentation masks guide the model to the salient patterns. A batch size of 32 is used for bitewings, and 128 for the tooth data. For Y-Net, we had to reduce the batch size to 16 and 32, respectively, due to memory constraints.

### C.4. Segmentation masks

Segmentation masks originally have the same dimension as the input image. In contrast, EMIL uses downscaled masks. To avoid that small carious lesions disappear due to downscaling, EMIL performs bilinear upsampling of the encoder output by a factor of 4 before extracting patches, resulting in spatial feature map resolutions of 64x84 for bitewing images and 48x48 for tooth images. Note that the primary task is classification, i.e., segmentation masks are used to guide training, but do not replace the classification label. A positive mask is only used if it corresponds to the class label; if the class label is negative, all elements of the mask are set to 0. Due to label noise, we do not use negative masks for the tooth-level task. Considering these filters, 35,683 masks ($\sim$97%) remain for the bitewing data, and 43,178 masks ($\sim$72% of all positive instances) remain for the tooth-level data. For more details on the performance of the caries segmentation model, see Cantu et al. (2020).

## Appendix D. EMIL hyperparameters

EMIL has two interesting hyperparameters, which we want to explain in more detail: $K_{min}$ and the patch size. Hyperparameter $K_{min}$ represents the minimum collective weight that must be assigned to the set of patches to be able to obtain a confident positive classification (i.e., $\hat{y} = 1$). For simplicity, consider the case where attention weights can only take on values in $\{0, 1\}$. Then $K_{min}$ can be thought of as the minimum number of patches that must be attended to. If this constraint is violated, the denominator of Eq. 4 turns into a constant, and the network is incentivized (for the positive class) to attend to more patches by increasing $\|\boldsymbol{w}\|$ through the nominator. Note that the value of $\hat{y}$ also depends on $\tilde{\boldsymbol{y}}$, i.e., attended patches must be classified positively to obtain a high positive class score. Fig. 5 shows the effect of $K_{min}$ on the patch weight map. For increasing values of $K_{min}$, sensitivity increases but precision decreases. If the value is too high, performance decreases because healthy tooth regions will be attended, which erroneously reduces disease probability (see, e.g., the first row of Fig. 5). When $K_{min} = 0$, little attention is assigned to any patch because all possible class scores can be obtained independent of $\|\boldsymbol{w}\|$. According to the standard

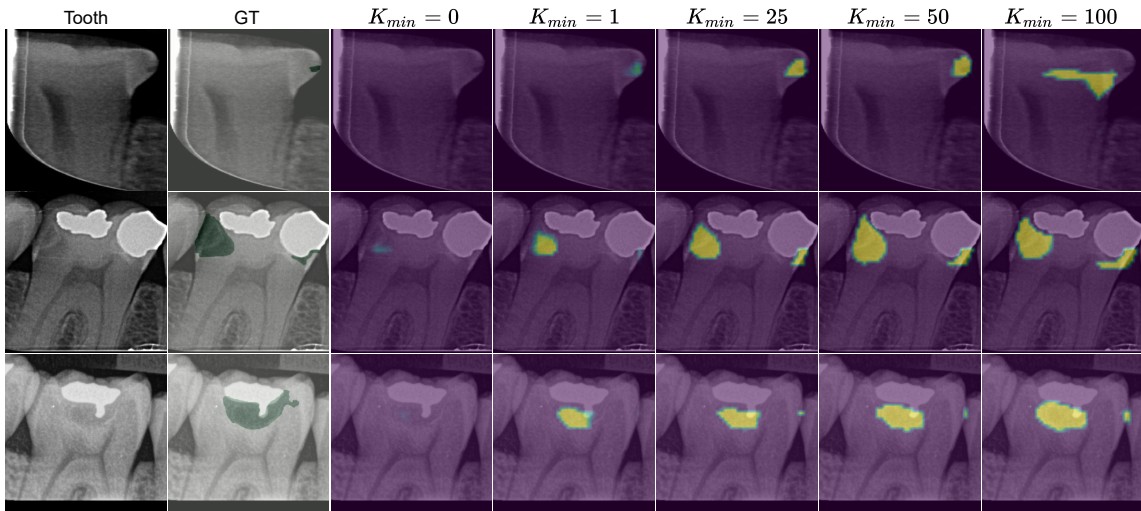

Figure 5: Weight maps for increasing $K_{min}$. Sensitivity increases, precision decreases.

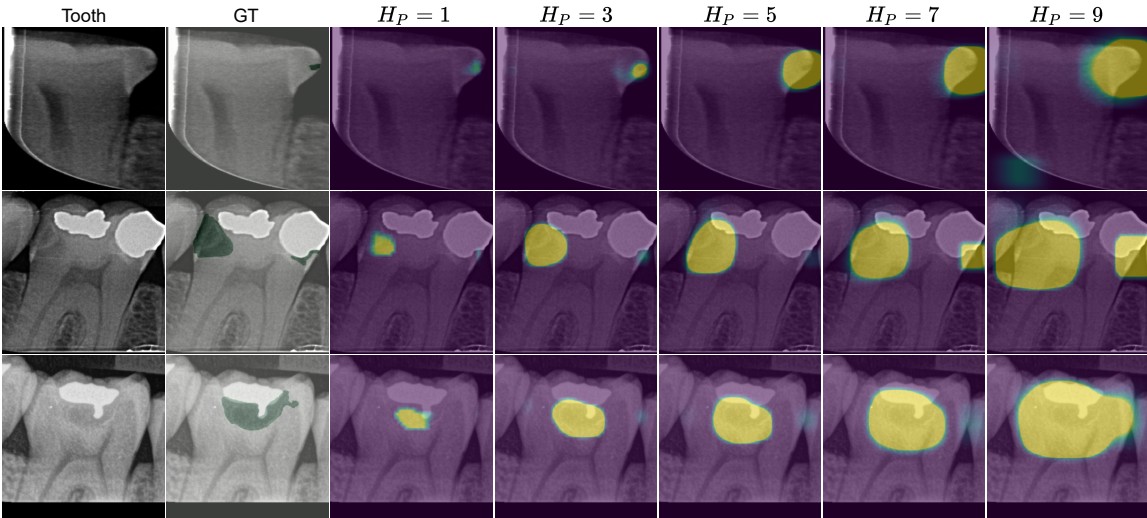

Figure 6: Unnormalized weight maps for increasing patch sizes, with $H_P = W_P$. Sensitivity increases, precision decreases.

MIL assumption (Dieterich et al., 1997; Foulds and Frank, 2010), a single positive instance is sufficient to positively label a bag, therefore $K_{min}$ is set to 1 and must not be searched.

The second hyperparameter is the patch size, which we set equal for both dimensions, $H_P = W_P$ – we use $H_P$ in the following to denote both width and height. The patch size controls the individual regions that are classified. If $H_P = H_U$, then a single global patch is considered and the training behavior is similar to a standard CNN. Fig. 6 shows the effect of $H_P$ on the heatmap for $H_S = 1$. Attention weights of overlapping patches are summed and clipped at 1 to improve visualizations. One can observe that the sensitivity increases while

precision decreases. In our experiments, the patch size had little impact on classification performance, but we prefer a small value for precise localization. Note that a small patch in the embedding space has a large receptive field and thus sufficient context to detect both small and larger lesions (Luo et al., 2016). For the main experiments, we set $K_{min} = 1$, $H_P = W_P = 1$ and $H_S = W_S = 1$.

## Appendix E. Computational comparison

Table 5 provides more computational details. Runtimes refer to a single forward + backward pass w/o data loading, averaged over 500 iterations with one warm-up iteration. Memory values refer to the peak GPU memory requirements during training. The parameter count refers to ResNet-18 as underlying encoder. One can observe that EMIL adds only little computational overhead compared to ResNet-18. In comparison, DeepMIL is considerably less efficient because overlapping patches are processed, which increases the effective input size. Furthermore, Y-Net is less efficient because an expensive decoder is used.

Table 5: Computational comparison for bitewing dataset, batch size 16.

|              | GFLOPs | #Params [M.] | Runtime [ms] | RAM [GB] |
|--------------|--------|--------------|--------------|----------|
| ResNet-18    | 23.9   | 11.2         | 44           | 2.48     |
| ResNet-50    | 55.4   | 23.5         | 143          | 9.19     |
| DeepMIL-32   | 90.8   | 11.3         | 256          | 9.43     |
| DeepMIL-128  | 80.4   | 11.3         | 133          | 7.51     |
| ViT          | 24.1   | 11.5         | 50           | 3.40     |
| EMIL         | 24.0   | 11.3         | 45           | 2.48     |
| Y-Net        | 248.2  | 17.8         | 318          | 23.54    |

## Appendix F. Further visualizations

Fig. 7-10 show further visualizations of true positive and false negative bitewings and teeth. When correct, both models detect lesions with few false positive visualizations. One reason for misclassification is low attention weights. For example, consider the first row of Fig. 8, where the patch prediction heatmap weakly highlights both lesions, however little attention is assigned to them. Nevertheless, a dentist may use these maps to detect caries and mark lesions so that the network can learn to locate them explicitly.

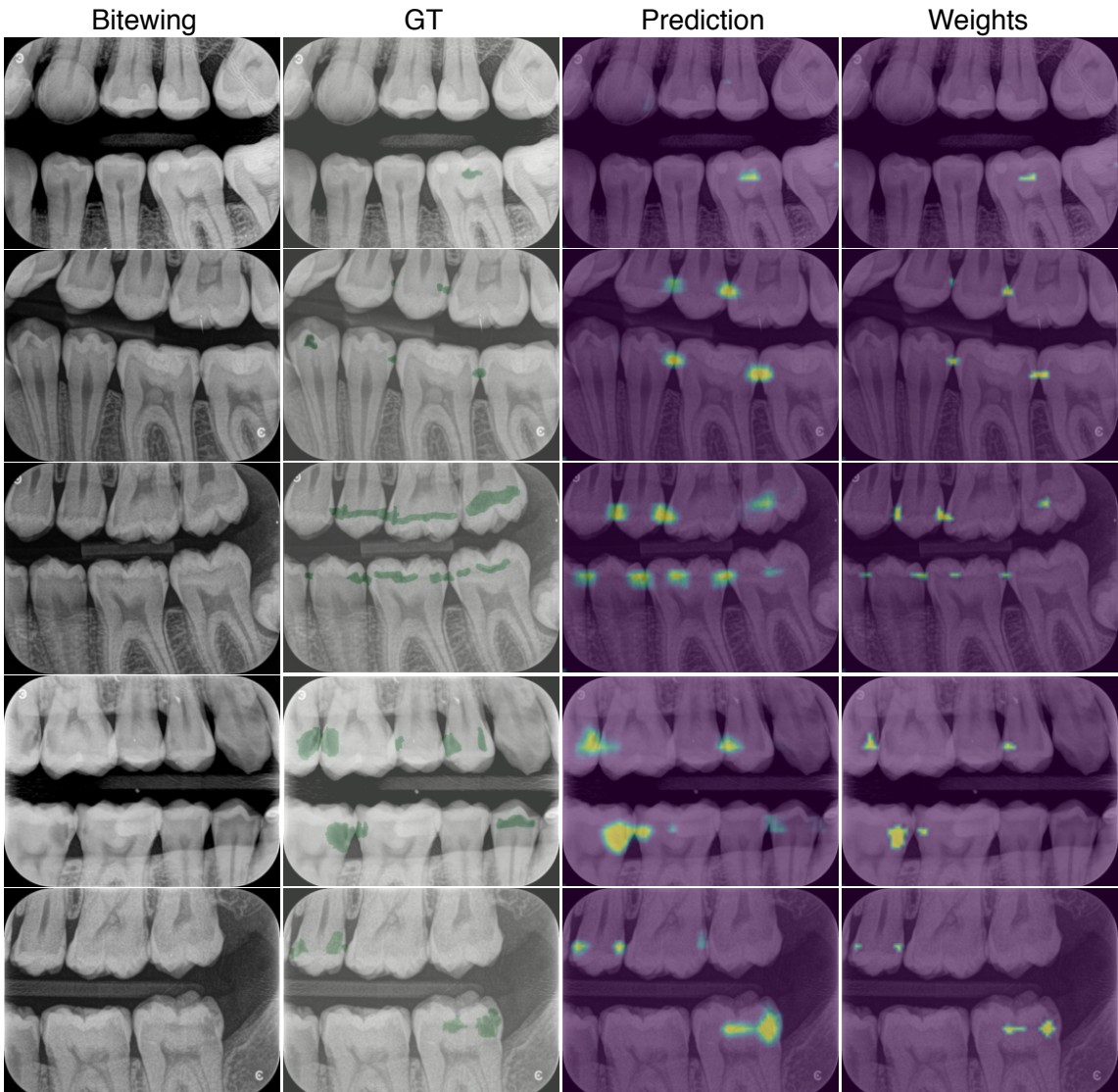

Figure 7: True positive bitewings and EMIL heatmaps. Trained w/o expert segmentations.

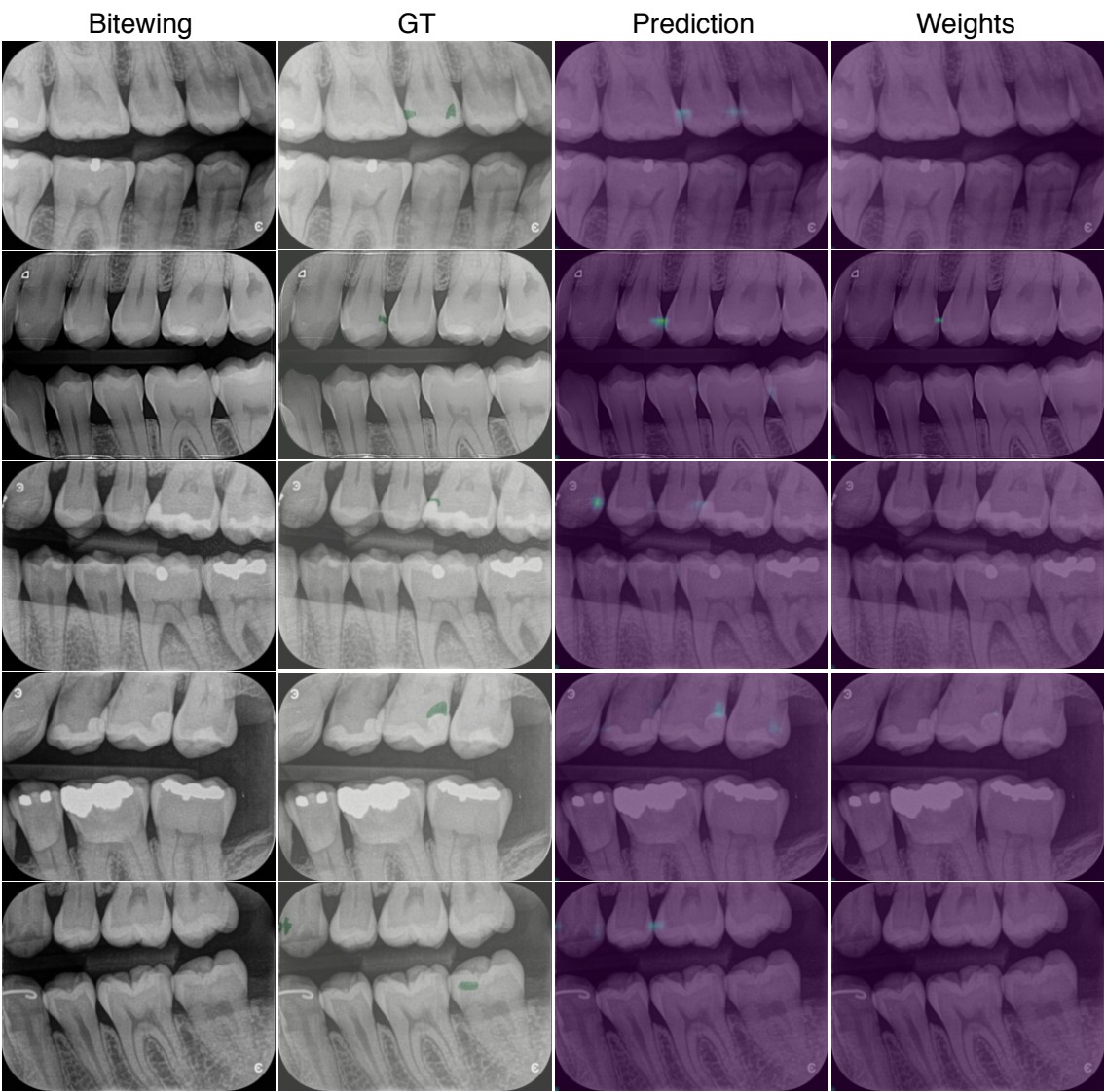

Figure 8: False negative bitewings and EMIL heatmaps.

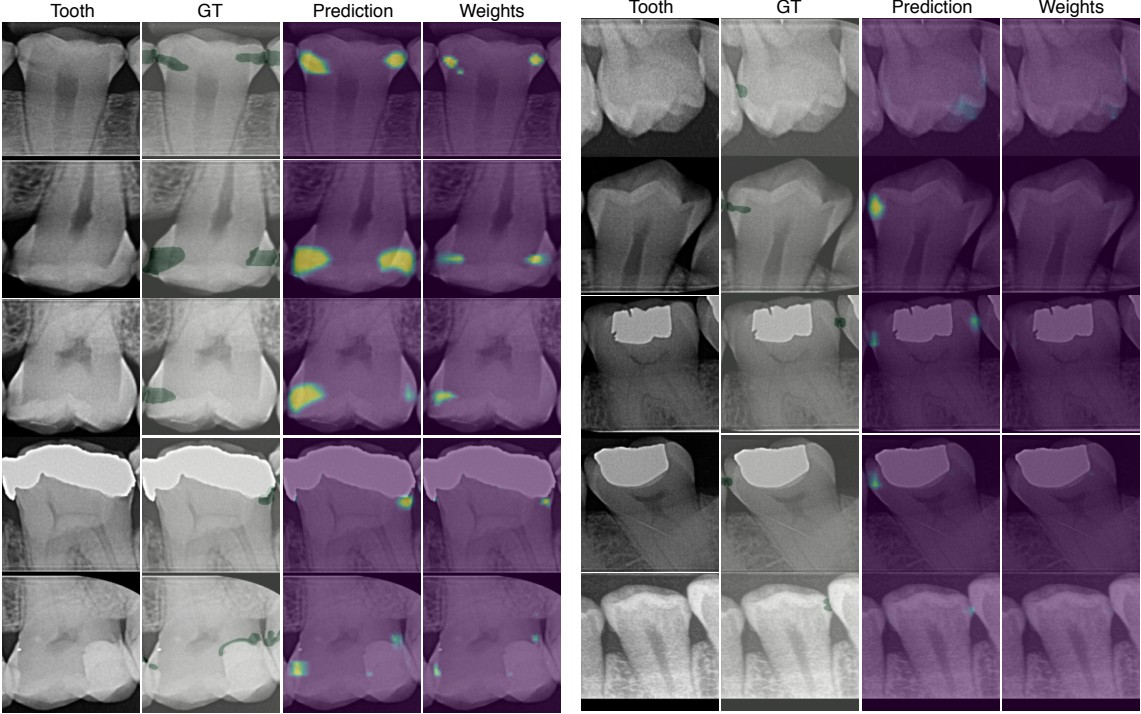

Figure 9: True positive teeth.          Figure 10: False negative teeth.

