# OpenReview forum: "Interpretable and Interactive Deep Multiple Instance Learning for Dental Caries Classification in Bitewing X-rays"
_MIDL.io/2022/Conference — MIDL 2022_

### Official Review · Reviewer_bgG2 · 2022-01-15

**Confidence:** 4
**Preliminary Rating:** 4
**Recommendation:** Poster

**Summary:**

This paper proposed an interpretable and interactive dental caries classification method in bitewing x-ray images. The deep multiple instance learning strategy and the attention mechanism are employed to design the classification model. Elaborate experiments conducted on a large in-house dataset show good performance.

**Strengths:**

1. The method is easy to follow.
2. The experiment is very informative.
3. The proposed method shows the potential for improvement in classification and localization performance by introducing external caries segmentation labels.

**Weaknesses:**

Some technical errors exist in the manuscript.
1. EMIL extracts "3D patches" from a spatial embedding resulting from any CNN. To my knowledge, the bitewing x-ray images are 2D data.
2. Table.2 ResNet-18 + caries segmentation model shows the best performance in Spec. metric in Tooth dataset.

**Deanonymize Review:**

no

**Final Rating After The Rebuttal:**

4: Weak Accept

**Justification Of The Final Rating:**

I read the author's reply and their response addressed my concern, I had no other additional comments. My final opinion is to agree to accept this paper, but if the end result was rejection, I could accept it.

**Paper Type:**

both

**Questions To Address In The Rebuttal:**

1. What caught my curiosity was on the tooth data and bitewing data settings in Table.1, when adding the caries segmentation model, why
did Y-Net and EMIL show completely opposite performances?
2. How many caries classes are involved? Each class metric or classification confusion matrix needs to be included.

**Special Issue:**

no

---

### Official Review · Reviewer_CSmP · 2022-01-25

**Confidence:** 5
**Preliminary Rating:** 2
**Recommendation:** Poster

**Summary:**

The paper presents an approach for caries detection. The proposed model achieves a maximum accuracy of 76% on the caries detection task and a maximum F1-score of 79%
There are 2 contributions:
1. A heatmap output of local patch classification probabilities.
2. It is amenable to learning from segmentation labels to guide training.

**Strengths:**

1. The interpretability and effort to implement a method that could give insights into the decisions made by the neural network is a strong point in this paper. Providing insight to the user about the decision made by the neural network is a desired feature in all machine learning applications.

**Weaknesses:**

1. Overall, the paper could be structured differently. It is hard to follow the implementation details and the final experiments that are performed.
2. There is no significant contribution or novelty to the original MIL implementation.
3. The performance achieved by the proposed method is far below from other reports in the literature.
4. Unfortunately, while the strong point of this paper is the interpretability, the contribution is limited.


**Deanonymize Review:**

no

**Detailed Comments:**

I found the Method section and notation hard to follow.

1. What is "any" convolutional backbone? Is this a pre-trained neural network without the final pooling layer? I'm assuming a pre-trained network that is used to extract features from each crop/patch using a sliding window approach.
The input image has dimensions Hx, Wx, Cx and after this step, it has dimensions Hu, Wu, Cu, then additional patches are extracted from U for the next classification step. Why not extract patches directly from the input image?
Please review the paper "Detection and diagnosis of dental caries using a deep learning-based convolutional neural network algorithm - Jae-Hong Lee". In this paper, the approach used a classification neural network and achieved an accuracy of ~89%. The work presented here is ~13% below. Even the work presented by Megalan Leo and Kalpalatha Reddy (which is referenced in the paper) achieves an accuracy of 87.6% in the classification task.

2. Patch weighting and aggregation:
The notation in equation 5, shouldn't the equation read w = sigmoid(tanh(A(P)) dot sigmoid(B(P)) c)? Since A and B are the multilayer perceptrons?
How does changing an activation function make the weights independent of each other and allow ignoring patches? The formulation derives from attention mechanisms (gated, Bahdanau,  attention, multi-head, etc.), during training, will encourage the model to assign a higher weight to patches containing caries and push down the weight of patches that do not. Training the MLP (which assigns weights to each crop) is an effective mechanism to enforce the model to ignore patches that do not contain a caries lesion.

3. Overall, the problem could be re-formulated as a feature extraction step followed by an attention mechanism step. It comes as a great surprise that the accuracy achieved by the proposed approach achieves only a maximum accuracy of 76.64%. and F-Score of  79.52% given how successful attention mechanisms have been for other deep learning tasks.

4. Interpretability:
The grad-cam approach could also be used with the MLPs that calculate the weights for each patch, i.e., instead of using the main network use the MLPs to generate the output heat maps for interpretability. The contribution to model interpretability is limited.

5. Interactivity:
While proposing an approach in which a user can interact with an AI could be a novelty, in this particular setting I don't see the benefit for such a feature. The outcome would be an increased number of False Negatives and False Positives which are absolutely not desired in a clinical setting. To test this out, simply feed random patches of non-caries lesions and see the answers of the trained model.

6. Experiment setup:
Rewrite: "Due to class imbalance, the balanced accuracy is used as stopping criterion" -> We compute weights to balance our majority and minority classes and use the accuracy as a stopping criterion.

Also, it is better to use the loss function as a stopping criterion rather than the accuracy. The loss is a measure of certainty, the accuracy measures how many are correctly classified.
Additionally, I would expect that the tooth instance segmentation and caries segmentation model bias the training. Are there at any point negative samples introduced in training? The output trained model will be sensitive to this input/interactions, what happens when negative samples are given to the trained model?

7. Baselines:
The "backbone" is introduced here which is ResNet-18 or is it ViT? This is confusing.
More details about the architecture are introduced here. It is hard to tell what is actually implemented at the end. It seems that all of the mentioned networks are used in the experiments.


**Final Rating After The Rebuttal:**

3: Borderline

**Justification Of The Final Rating:**

Some of my concerns were addressed in the rebuttal and I found using a larger dataset to be convincing. It would be interesting to see a comparison with the method that has superior performance but has a cleaner dataset.

**Paper Type:**

both

**Questions To Address In The Rebuttal:**

1. Why is the interactivity of the algorithm a feature of this work? I would like to see a test by giving the trained model negative classes as input. What would be the decision of the model be if this is the case? Would it contradict the user inputs?
2. The proposed architecture is a concern. It has to be demonstrated that this architecture is far superior to using raw patches as input and training a neural network end-to-end. From the results, the architecture gains 1% accuracy which is not significant. It seems that hyperparameter tunning on ResNet-50 could outperform the proposed model.
3. Is RAM memory a concern? Will this model run in a low-resource environment?
4. Please review the detailed comments.

**Special Issue:**

no

---

### Official Review · Reviewer_6SCy · 2022-01-25

**Confidence:** 4
**Preliminary Rating:** 4
**Recommendation:** Poster

**Summary:**

This paper proposes a simple and efficient image classification architecture based on deep multiple instance learning (DIL), and firstly applies it to the challenging task of caries detection in dental radiographs. The proposed method outputs a heatmap of
local patch classification probabilities though trained with image-level labels. Also,  the method could be further refined by external caries segmentation models. Furthermore, the human user can faithfully interpret predictions and interact with the model to decide which regions to attend to. Extensive experiments have been conducted to demonstrate the efficacy of the proposed method.


**Strengths:**

1. The paper is well-written and adequately addresses prior works.

2. This paper presents an interpretable and interactive method named Embedding Multiple Instance Learning (EMIL), which can be trained with image-level labels, and be further refined with external caries segmentation models.

3. Extensive experiments have been conducted, and the experimental results are promising.


**Weaknesses:**

The paper claims the proposed EMIL is 6.6$\times$ faster than Y-Net, while consuming up to 9.5$\times$ less memory at similar/better mean performance. It will be more solid if the paper could provide more numbers/analysis to support this claim.

**Deanonymize Review:**

no

**Detailed Comments:**

From Tab.2, we can find that in the interactive settings, when considering all confidences (IoU@0), the localization performance of the proposed EMIL+ is inferior to the baseline Y-Net+. Any possible reasons or explanations for this? I would recommend the authors to dig deeper for these results.

**Final Rating After The Rebuttal:**

4: Weak Accept

**Justification Of The Final Rating:**

Thanks for the author response. The authors addressed most of my initial concerns. Also, considering all the other reviews and author respnses, I'd like to keep my original rating and recommend to accept this manuscript though there are small issues.



**Paper Type:**

methodological development

**Questions To Address In The Rebuttal:**

1. It will be more solid if the paper could provide more numbers/analysis to support the claims in terms of efficiency .

2. The analysis of poor localization performance in the interactive setting when compared with baseline.

**Special Issue:**

no

---

### Meta-Review · Area_Chair_GMzz · 2022-02-17

**Recommendation:** Accept (Poster)
**Confidence:** 4

**Metareview:**

This paper proposed a simple and efficient image classification model based on MIL. The experiments have been conducted on a large clinical dataset (~38K images). Although some issues regarding confusion of implementation details and methodological contribution have been raised by reviewers, based on the consensus of reviewers and merits on the application, a decision a accept is recommended.

---

### Decision · Program_Chairs · 2022-02-28

Accept